# Knowledge and Treatment of Asymptomatic Hyperuricemia Versus Gout Among Physicians in Saudi Arabia: A Cross-Sectional Survey

**DOI:** 10.3390/healthcare13212719

**Published:** 2025-10-27

**Authors:** Yousef M. Alammari, Abdulmohsen Albassam, Mohammad Alorainy, Faisal Alibrahim, Abdulrahman Alshahwan, Abdullah Alaskar, Rayan A. Qutob, Mohammad Alhajery, Abdulwahed Alotay, Yassir Daghistani, Abdulrahman Alanazi, Ibrahim Alshehri

**Affiliations:** 1Department of Internal Medicine, College of Medicine, Imam Mohammad Ibn Saud Islamic University, Riyadh 11623, Saudi Arabia; raqutob@imamu.edu.sa (R.A.Q.); maalhajery@imamu.edu.sa (M.A.); aaalotay@imamu.edu.sa (A.A.); amalanazi@imamu.edu.sa (A.A.); 2College of Medicine, Imam Mohammad Ibn Saud Islamic University, Riyadh 11623, Saudi Arabia; 441016903@sm.imamu.edu.sa (M.A.); 441024160@sm.imamu.edu.sa (F.A.); 441024371@sm.imamu.edu.sa (A.A.); 441017402@sm.imamu.edu.sa (A.A.); 441013558@sm.imamu.edu.sa (I.A.); 3Department of Medicine, Faculty of Medicine, University of Jeddah, Jeddah 23218, Saudi Arabia; ydaghistani@uj.edu.sa

**Keywords:** asymptomatic hyperuricemia, gout, knowledge, practice, physician

## Abstract

Background: Gout and asymptomatic hyperuricemia (AH) are common conditions for elevated levels of uric acid in the blood. While gout is a well-known and widely recognized condition, AH may be less familiar to healthcare professionals. This study aimed to estimate and determine the knowledge and treatment of AH versus gout among physicians in Saudi Arabia. Methods: A standardized, online validated questionnaire was used to collect data from physicians in Saudi Arabia. The tool comprised two parts: a 3-item section on demographics and a 37-item section assessing knowledge (18 items) and practice (19 items) related to the management of AH and gout. Convenience sampling was employed for participant recruitment. Data were analyzed using descriptive and inferential statistics, and *p* < 0.05 was considered statistically significant. The questionnaire was validated by three experts in rheumatology and clinical pharmacy, with a KR-20 of 0.87. Results: Of the 744 participants, 53.9% were female, and 59.1% were aged 24–34. A total of 58.7% had attended continuing medical education (CME) on AH or gout during the last three years. The mean practice score was significantly higher among residents compared to physicians without specialty training (mean difference = −1.43632, 95 %CI: −2.4575–−0.4151, *p* < 0.001) and consultants compared to physicians without specialty training (mean difference = −3.2769, 95% CI: −4.7918–−1.7620, *p* < 0.001). Male physicians scored higher than female physicians (1.73 ± 1.08 vs. 1.46 ± 1.09, *p* = 0.001). Consultants and orthopedic specialists demonstrated the highest knowledge levels (*p* < 0.001). Conclusion: The knowledge and practice of physicians for managing AH or gout were unsatisfactory. Younger female general practitioners were more likely to exhibit poor knowledge and practice in managing AH or gout cases. Despite advanced diagnostic tools and treatment processes, physicians have many misconceptions. Hence, continuous medical education focusing on AH or gout is vital to address these misconceptions.

## 1. Introduction

Gout and asymptomatic hyperuricemia (AH) are common conditions related to elevated levels of uric acid in the blood. While gout is a well-known and widely recognized condition, AH may be less familiar to healthcare professionals. AH is defined as a blood serum uric acid concentration exceeding 7 mg/dL for adult males and 6 mg/dL for adult females [1,2]. A total of 90% of individuals with hyperuricemia have AH, indicating that only a small percentage of hyperuricemia patients have gout.

Nevertheless, AH is thought to represent the initial stage of advanced gout and can exist for more than 10 years prior to flare-ups [3]. Understanding the knowledge and perspectives of physicians and pharmacists regarding these conditions is crucial for effective management and patient education. Although extensive research has been conducted on gout and hyperuricemia in many countries, limited information is available regarding the knowledge and treatment of physicians in Saudi Arabia. The Saudi healthcare system plays a vital role in managing and providing healthcare services to diverse populations. Globally, gout prevalence ranges from less than 1% to 6.8% of the population depending on the research methodology and the population group under study [4]. Major risk factors for hyperuricemia and gut include male gender, older age, obesity, hypertension, chronic kidney disease, metabolic syndrome, and excessive intake of red meat, sea food, and alcohol [4]. Few epidemiological studies on gout and asymptomatic hyperuricemia (AH) exist in Saudi Arabia. Al-Arfaj et al. estimated the prevalence of hyperuricemia in Saudi Arabia as 8.4% in 2001 [1]. Mohrag et al. reported that 14% of 1208 participants had gout in 2022 [5], while a recent study reported that 17.3% of Saudi adults had gout [6]. This indicates that more research should be conducted on such conditions in Saudi Arabia. However, recent data evaluating physicians’ knowledge and clinical practices regarding these conditions in Saudi Arabia remains limited. This represents a significant knowledge gap that may affect patient management and outcomes. Therefore, it is essential to assess the knowledge and treatment of physicians in Saudi Arabia regarding AH and gout.

Physicians play a vital role in the diagnosis and management of AH and gout. Inadequate knowledge or misconceptions may lead to underdiagnosis, delayed treatment, and poor disease control [4,7]. This study hypothesized that physicians’ socioeconomic and practice characteristics will affect their knowledge and treatment of asymptomatic hyperuricemia versus gout. This study aimed to assess the knowledge and treatment of AH and gout among physicians in Saudi Arabia. The findings of this study can contribute to the development of targeted educational interventions and guidelines to improve the management of AH and gout. Enhancing patient education and awareness about these conditions can also be facilitated by identifying knowledge gaps and misconceptions among physicians.

## 2. Materials and Methods

### 2.1. Study Design and Setting

A cross-sectional descriptive study was conducted among physicians residing in Saudi Arabia to evaluate their knowledge and practice regarding AH and gout. Data collection was conducted using a self-administered online questionnaire distributed via social media platforms from June 2024 to February 2025.

### 2.2. Sampling Technique and Participants

The study population consisted of physicians certified by the Saudi Commission for Health Specialties, currently residing in Saudi Arabia, regardless of specialty. Based on statistics published by the Ministry of Health, the total number of physicians in Saudi Arabia is approximately 113,300 [8]. Using a 95% confidence level and a 5% margin of error, which are the most often used in biomedical research [9], the minimum calculated sample size required for the study was 383 participants using Raosoft online sample size calculator. Convenience sampling was employed for participant recruitment. All physicians meeting the inclusion criteria (certified by the Saudi Commission for Health Specialties and residing in Saudi Arabia [10] were eligible for participation, with no further exclusion criteria applied once the inclusion criteria were satisfied. Exclusion criteria were non physicians, interns, pediatricians and other physicians not treating AH or gout.

### 2.3. Definition of General Practitioner and Physician Without Specialty Training

In this study, physicians were classified according to the categories used in the Saudi healthcare context:

General Practitioner (GP): A medical doctor who has completed a medical degree and the mandatory one-year rotating internship, but has not enrolled in any specialty training (residency) program. These physicians most often work in primary healthcare centers and provide general medical services.

Physician without specialty training: A medical doctor who has likewise completed medical school and internship and is practicing in a clinical field such as internal medicine, general surgery, orthopedics, or other specialties, but who has not entered or completed a formal residency or specialty training program. These physicians may provide inpatient or outpatient care within these fields, but do not hold specialist certification.

### 2.4. Study Measurements

The questionnaire utilized in this study was developed based on a previously published study conducted in the Qassim region of Saudi Arabia [11]. We revised and added some questions and answers to the practice section of the questionnaire to comply with the 2020 American College of Rheumatology guidelines [12]. Furthermore, we adjusted the sociodemographic section of the questionnaire for consistency with the study’s aim and included different medical specialties and various levels of medical training/practice.

Content validity was confirmed by a panel of three experts in rheumatology and clinical pharmacy, who reviewed the items for clarity and relevance. A pilot study was conducted among 30 physicians to assess clarity and reliability. Based on feedback, minor wording modifications were made to improve question clarity. Reliability testing showed a good internal consistency with a KR-20 value of 0.87 for the total practice score. It included demographic information, such as gender, age, specialty, years of clinical experience, and level of training. Physicians’ knowledge and practices regarding AH and gout were assessed using structured questions, as outlined below in the questionnaire criteria.

### 2.5. Questionnaire Criteria

The questionnaire consisted of two main sections: a knowledge section (3 items of multiple-choice question (MCQ) format) and a practice section (37 items of MCQ format). The knowledge of AH and gout was assessed using a three-item questionnaire; the correct answer for each question was coded with 1, while the incorrect answer was coded with 0. The total knowledge score was calculated by adding all three items, achieving scores ranging from 0 to 3 points. The higher the score, the greater the knowledge of AH and gout. Physicians were classified as having poor knowledge if the score was between 0 and 1, a score of 2 was considered moderate, and a score of 3 was considered a good knowledge level.

Likewise, the practice toward the treatment of AH and gout was assessed using a 37-item questionnaire, with the correct answer for each identified and coded with 1, while the incorrect answer was coded with 0. The total practice score was calculated by adding all 37 items. Scores ranging from 0 to 37 points were generated. The higher the score, the higher the practice toward the treatment of AH and gout. To determine the level of practice, physicians were considered to have poor practice if their score was less than 50%, moderate with scores of 50% to 75%, and a good practice level for scores above 75% [11]. A poor practice leads to poor management of patients, moderate leads to moderate management of patients, and good practice leads to good management of patients with AH and gout. The ACR 2012 non-pharmacologic, ACR 2020 guideline, EULAR/BSR lifestyle recommendations) were used to define “correct” responses for practice and diet items a priori. We operationalized “AVOID” for items that the guidelines recommended to avoid or strongly discourage (such as sugar-sweetened beverages, organ meats, and alcohol overuse) and “LIMIT” for items that were recommended to be consumed in small amounts or reduced (such as most seafood, red meat, small amounts of alcohol, and food with added sugar/salt). Vegetables and low-fat dairy were classified as “neutral or encouraged”. Appendix A contains the item-to-guideline mapping. The first knowledge item assessed the threshold for serum uric acid defining asymptomatic hyperuricemia. The correct answer was predefined as >7 mg/dL for men and >6 mg/dL for women based on the Al-Arfaj nationwide study, Du, L. et al., and US NHANES [1,2,13]. Other guidelines suggested slightly different thresholds (6.8 mg/dL for both sexes), which we acknowledge as a source of heterogeneity. Because the question was multiple choice with a single correct answer, alternative threshold could not be applied to participant responses.

### 2.6. Statistical Analysis

For the ease of analysis and comparison, certain categories in demographic variables were merged based on their types and similarities. This pattern is necessary to achieve statistically significant results, which may be deemed fit for the study.

The mean and standard deviation were used for the descriptive analysis of the metric variables, while frequency and proportion (%) were given for all categorical variables. The difference between knowledge and practice scores among the sociodemographic characteristics of the physicians was performed using the Mann–Whitney Z-test and the Kruskal–Wallis H-test. Furthermore, post hoc analyses were conducted to determine the multiple mean differences in knowledge and practice scores in relation to the level of training using the Dunn–Bonferroni test. The Spearman correlation coefficient was also used to determine the correlation between knowledge and practice scores. A normality test was performed using the Kolmogorov–Smirnov test. Based on the plots and statistics, the knowledge and practice scores follow a non-normal distribution (*p* < 0.001), and the knowledge score skewness and kurtosis were −0.28 ± 0.09 and −0.94 ± 0.18, respectively. Similarly, for the total practice score with a positive skewness (0.42 ± 0.09) and negative kurtosis (−0.42 ± 0.18). Together with the observed skewness and kurtosis values outside the range of ±1, these results suggest that both variables were not normally distributed Therefore, the non-parametric tests were applied. This means that the non-normal distribution of the scores allowed non-parametric tests to be conducted. However, if the scores followed a normal distribution, parametric tests would be applied.

Values were considered significant at the *p* < 0.05 level. All data analyses were performed using Statistical Package for the Social Sciences (SPSS) version 26 software (IBM Corporation, Armonk, New York, NY, USA).

### 2.7. Ethical Considerations

The informed consent document indicated the purpose of the study and the rights of the participant for confidentiality. It also explained that they could withdraw at any moment with no commitment to the study team. The online survey was designed to ensure confidentiality and data integrity. IP addresses were not recorded, and measures were implemented to prevent duplicate submissions. All responses were stored in secure, password-protected database accessible only to the study team.

Ethical approval for this research was obtained from the Institutional Review Board (IRB) in Imam Mohammad Ibn Saud Islamic University (Registration: HAPO-01-R-0011, Project number 659, Approval date: 20 June 2024).

## 3. Results

### 3.1. Physician Demographics

In total, 744 physicians responded to our survey. Table 1 presents the sociodemographic characteristics of the physicians. Of the total, 59.1% were between 24 and 34 years old. Female physicians (53.9%) were predominantly higher in numbers than male physicians (46.1%). Physicians who lived in the Western Region constituted 46.2%. The most common medical specialty was general practitioner (31.3%). And the most common level of medical training was physician without specialty training (41.8%). Physicians who had less than five years in practice were 51.5%. In addition, 58.7% had attended continuing medical education (CME) activities related to AH, 66.8% actively sought out and reviewed information about the disease, and 58.6% were aware of the 2020 American College of Rheumatology guidelines for managing gout [10].

### 3.2. General Knowledge and Practice

Regarding the assessment of the knowledge about AH and gout (Table 2), nearly half (48.9%) knew that the correct threshold for serum uric acid levels defining AH was >7 mg/dL in males and >6 mg/dL in females, and 54.7% knew that AH does not always progresses to gouty arthritis, whereas 55.2% were aware that AH does not always require treatment. Based on the knowledge categories mentioned above in the questionnaire criteria and statistical analysis, the total mean knowledge score was 1.59 (SD 1.09), and the median was 1.0 and the IQR was 1, with poor, moderate, and good knowledge constituting 48.3%, 24.5%, and 27.2%, respectively (see also Figure 1A). Regarding the practice, 50.3% knew the most appropriate method to perform during the initial evaluation of a patient with a suspected gouty attack.

However, only 28.7% knew the correct approach to initiate treatment for a patient with acute gout flare. In addition, only 31.3% considered any of the abortive anti-inflammatory medications as a first line of therapy based on the clinical situation. Physicians who knew the duration of anti-inflammatory prophylaxis to be continued following ULT for gout were only 29.7%. The most common condition in which physicians would typically initiate ULT was the incidence of more than one gout per year (54.2%). 52.7% were correct that 100 mg once daily and titrating up was the starting dose for initiating allopurinol for ULT in patients with gout. In addition, 46.5% knew the target range for serum uric acid levels after starting ULT for gout. 59.9% were correct that increasing the dose of the same ULT or switching to another ULT would be appropriate if the target serum uric acid range had not been achieved after 3–6 weeks of ULT. Only 33.3% knew indefinite ULT with regular follow-up uric acid levels for patients with gout, and nearly half of them (49.5%) were aware that routinely discussing the importance of diet and lifestyle modifications with patients and providing resources or referrals is needed in clinical practice.

According to physicians, the most common diet to avoid for patients with gout was organ meats high in purine content (58.7%), while the most common diet to limit was beef or lamb (42.5%). Based on the above practice items, the total mean practice score was 19.9 (SD 4.16), and the median was 19.0 and the IQR was 5. Accordingly, 44.8%, 51.0%, and 4.2% were considered to have poor, moderate, and good practice levels, respectively (see Figure 1B). To examine the robustness of the findings, a sensitivity analysis was performed using alternative cutoffs based on the quartile distribution of practice scores (≤17 = poor, 18–22 = moderate, ≥23 = good). Using these thresholds, 37.8% of participants were categorized as having poor practice, 39.7% as moderate, and 22.6% as good practice. The overall results and group comparisons remained consistent, confirming the stability of the conclusions. In addition, 7.4% (n = 55) of physicians did not know the answers to all the questions in the knowledge section.

### 3.3. Knowledge and Demographics

Measuring the association between knowledge scores and the sociodemographic characteristics of the physicians found that higher knowledge scores were associated with increasing age (Z = 4.518; *p* < 0.001), male physicians (Z = 3.380; *p* = 0.001), increasing years of experience (Z = 4.417; *p* < 0.001), and actively seeking and reviewing information focused on AH (Z = 3.997; *p* < 0.001). No significant differences were observed between the knowledge scores in relation to attendance to CME activity focusing on AH and awareness of the latest 2020 American College of Rheumatology guidelines for the management of gout (*p* > 0.05). Regarding medical specialties, all the physicians were combined in their specific medical specialty, regardless of their level of medical training. The highest knowledge scores were seen in the total of all orthopedic surgeons (mean score: 2.05), followed by family medicine physicians (mean score: 1.78), and internal medicine (mean score: 1.72). General practitioners showed the lowest scores (mean score: 1.33), with significant differences across the groups (H = 34.403; *p* < 0.001). As for levels of medical training/practice (regardless of the physician’s medical specialty), consultants, who are at the highest level of medical training/practice, showed the highest knowledge score (mean score: 2.12), followed by specialists/registrars (mean score: 1.81) and residents (mean score: 1.65). Fellows showed the lowest knowledge score (mean score: 1.35), with significant differences across the groups (H = 31.791; *p* < 0.001). (Table 3).

### 3.4. Practice and Demographics

Examining the association between practice scores and the sociodemographic characteristics of the physicians revealed that higher practice scores were associated with increasing age (Z = 5.643; *p* < 0.001), male gender (Z = 2.882; *p* = 0.004), increasing years of experience (Z = 4.172; *p* < 0.001), not attending CME focusing on AH (Z = 3.353; *p* = 0.001), actively seeking and reviewing information about AH (Z = 3.267; *p* = 0.001), and familiarity with the latest 2020 American College of Rheumatology guidelines for the management of gout (Z = 2.017; *p* = 0.044). Regarding medical specialty, by combining all the physicians in their specific medical specialty regardless of their level of medical training, the highest practice scores were seen in orthopedic surgeons (mean score: 21.3), followed by family medicine physicians (mean score: 20.6), and internal medicine physicians (mean score: 20.4). At the same time, general practitioners showed the lowest scores (mean score: 18.7), with significant differences across the groups (H = 26.529; *p* < 0.001). As for levels of medical training/practice (regardless of which medical specialty the physician is in), consultants showed the highest practice score (mean score: 21.3), followed by specialists/registrars (mean score: 20.7), and residents (mean score: 20.4). At the same time, physicians without specialty training showed the lowest score (mean score: 18.9), with significant differences across the groups (H = 45.425; *p* < 0.001). (Table 4).

### 3.5. Knowledge and Practice Level

Table 5 shows the multiple mean differences in the knowledge scores in relation to the level of medical practice. It was observed that there was a statistically significant difference in mean knowledge score between physicians without specialty training and specialists/registrars (*p* = 0.021), between physicians without specialty training and consultants (*p* < 0.001), and between fellows and consultants (*p* < 0.001). These imply that physicians without specialty training have very low knowledge scores when compared to specialists/registrars and consultants. This also implies that fellows have extremely low knowledge scores when compared to consultants.

### 3.6. Practice Scores and Practice Level

In Table 6, multiple mean differences of practice score in relation to the medical practice indicated that there was a statistically significant difference in the mean practice score between physicians without specialty training and residents (*p* = 0.001), between physicians without specialty training and specialists/registrars (*p* = 0.003), physicians without specialty training versus consultants (*p* < 0.001), and fellows versus consultants (*p* < 0.001). These imply that physicians without specialty training have very low practice scores when compared to residents, specialist/registrars, and consultants. This also implies that fellows have very low practice scores when compared to consultants.

### 3.7. Knowledge and Practice Scores and Specialties

Table 7 shows the knowledge and practice scores of various levels of medical training/practice across the medical specialties. The residents (N: 18) in orthopedics in our study had the highest knowledge score (mean score: 2.56), followed by specialists/registrars (N: 23) in internal medicine (mean score: 2.39), and consultants (N: 21) in internal medicine (mean score: 2.33). The consultants (N: 10) in rheumatology, though having a low knowledge score (mean score: 1.60), had the highest practice score (mean score: 24.6), followed by residents (N: 18) in orthopedics (mean score: 23.2), and consultants (N: 21) in internal medicine (mean score: 22.7).

Adjusted for age, gender, years in practice, specialty role, and guidelines awareness, multiple linear regression indicates that CME attendance was significantly associated with higher practice (β = 1.368; 95% CI = 0.738–1.998; *p* < 0.001) (Table 8).

Figure 2 illustrates a significant positive correlation between knowledge and practice scores (rs = 0.496; *p* < 0.001), indicating that an increase in knowledge score is associated with a corresponding increase in practice score.

## 4. Discussion

This study explores physicians’ knowledge and practices in managing patients with AH or gout. Research in this discipline is vital for understanding how gout and hyperuricemia are managed, ultimately leading to improved healthcare delivery and patient outcomes.

### 4.1. Knowledge About AH

The physicians’ knowledge regarding AH was unfavorable. Based on the three items representing knowledge, the overall mean knowledge score was 1.59 out of 3 points: 48.3% had poor knowledge, and only 27.3% were deemed good. This is consistent with the study by Alqarni and Hassan [14], which reported inadequate knowledge levels among primary healthcare physicians. Contradicting these reports, several studies documented better physicians’ knowledge of the disease [15,16,17]. These differences could be due to diverse educational backgrounds, evolving guidelines, regional variations, access to continuing education, and research methodologies.

Our findings were consistent with the results of Alraqibah et al. [11]. Of the PHPs, 32.3% attended CME on AH or gout, whereas 67.7% knew of the guidelines for managing these diseases. Additionally, they noted that more experience and CME attendance on AH and gout contributed to better understanding and practice levels [11]. Our findings and those of Alraqibah et al. [11] and Tawhari et al. [16] found that attendance in CME on AH or gout and reading topics related to these diseases increases knowledge and practice. However, these results did not achieve statistical significance (*p* > 0.05). The continuous pursuit of education and awareness of guidelines equip physicians with the necessary tools and knowledge to effectively manage gout, benefiting both their practice and their patients.

The lack of a physician’s understanding of AH was evident in the specific knowledge item. For instance, despite more than half (55.2%) being correct that AH does not always need treatment, with a similar proportion (54.7%) knowing that AH does not always lead to gouty arthritis, their understanding of the appropriate threshold for serum uric acid (SUA) levels seemed to be low (48.9%). These findings do not agree with those of Alenazi et al. They found that family medicine residents understood the correct values for SUA-defining AH better (61.5%), and most residents (70.9%) disagreed that AH always progresses to gouty arthritis, while a similar proportion believed that AH may not necessarily need treatment [18]. Regarding physicians’ knowledge and management of gout, Sautner and Sautner [19] found that most physicians (86%) knew of the protocol for the SUA target, and an even higher proportion of physicians (96.1%) indicated a therapy change just in case of missing this target.

According to Rai et al. [20], four themes were identified: limited knowledge about gout, healthcare providers’ interactions, experiences, and attitudes toward taking medication, and practical barriers to long-term use of drugs. Strategies to address the gaps in knowledge are necessary to improve patient outcomes. A multifaceted approach involving education, communication, and resource access may enhance the physician’s perspective of the disease.

### 4.2. Significant Factors Contributing to Knowledge

The results of this study concluded that age over 35 years, consultants, orthopedic surgeons, and years of experience are significant predictors of increased knowledge. Older physicians with more years of experience often have more clinical exposure than younger physicians with fewer years of experience. They better understood clinical guidelines, practice-based mentorship, and exhibited confidence in clinical judgment. Likewise, the orthopedic surgeon’s expertise is rooted in their direct, firsthand experience with the joint complications of gout and hyperuricemia, and ability to manage the full spectrum of musculoskeletal manifestations, distinguishing them from other specialties that may focus more on systemic or medical management. This is consistent with the study by Alraqibah et al. [11], suggesting that primary healthcare providers (PHPs) over 45 years old and those with more than 10 years of experience were associated with higher knowledge scores.

In contrast to these reports, Tawhari et al. [16] found no significant differences in knowledge scores according to age, gender, and years of experience (*p* > 0.05). Supporting these reports, Tiwaskar and Sholapuri [17] also found no significant variation in knowledge between male and female physicians. In our study, we found higher knowledge scores were associated with increasing age (*p* < 0.001), male physicians (*p* = 0.001), increasing years of experience (*p* < 0.001), and actively seeking and reviewing information focused on AH (*p* < 0.001).

Additionally, post hoc analysis implied that physicians without specialty training may have the poorest knowledge of AH and gout than other physicians. The combination of a broader training focus, limited experience with complex cases, less exposure to specialized guidelines, and fewer opportunities for targeted continuing education likely contributes to the relatively poorer knowledge of AH and gout among physicians without specialty training and fellows compared to their more senior or specialized counterparts.

### 4.3. Low Knowledge Scores of Rheumatologists

Despite consultants in rheumatology achieving the highest *practice* score (mean score: 24.6) across different specialties in this study, they generally had a low *knowledge* score (mean score: 1.35). That is because the answer most often chosen to the first question in the knowledge section, as depicted in Figure 3, was: The threshold for SUA levels defining AH had a SUA level >6 mg/dL in both males and females. We did not consider this answer correct because a nationwide study conducted in Saudi Arabia and many other studies, such as the U.S. National Health and Nutrition Examination Survey (NHANES), defined AH as serum urate (SU) greater than 7 mg/dL for males and greater than 6 mg/dL for females, which we considered to be the correct answer for our study [1,2,21]. Furthermore, the response did not reflect the latest American College of Rheumatology guidelines definition of AH [12]. However, 24 (44.4%) out of 54 rheumatologists selected this response most often. Half of the 24 rheumatologists who chose this answer had not attended any CME activities focused on AH or gout.

This is understandable because there are different guidelines under which the answers would be correct. The 2012 American College of Rheumatology guidelines defined hyperuricemia as SU greater than either 6.8 or 7.0 mg/dL for both genders [22]. The 2020 American College of Rheumatology guidelines defined AH as 6.8 mg/dL for both genders [12]. Other studies, such as Bardin et al. [3] suggest the definition of AH to be SU greater than 6 mg/dL for both genders, reflecting the correct answers of the 24 rheumatologists. Therefore, these differences in the guidelines and studies defining AH might have affected the responses and the correct answer. Also, the reason why the 24 rheumatologists in our survey chose this answer may have been that the 2020 American College of Rheumatology guidelines recommend achieving and maintaining an SU target of <6 mg/dL for both male and female patients [12].

### 4.4. Practice Related to AH or Gout

In our study, physicians showed inadequate practices in managing AH and gout. According to the given criteria, almost half of the respondents (44.8%) were likely to have poor practices in managing patients with AH or gout, while 55.3% had moderate to good practice levels. This is comparable to the reports by Alraqibah et al. [11] suggesting that one-fourth of primary care providers had poor practices in managing AH and gout cases. In contrast, various studies have documented improved practices among practicing physicians in managing such conditions [15,16,17,18].

### 4.5. Significant Factors in Practice

All demographic variables were strongly associated with practice scores (*p* < 0.05) as shown in Table 3 and Table 4. In particular, higher practice levels were associated with older age groups, male physicians, orthopedic surgeons, or consultants, and years of experience. Furthermore, orthopedic surgeons who are consultants may be associated with more specialized knowledge and experience in musculoskeletal disorders, a focus on functional outcomes, experience with complex cases, and better interdisciplinary collaboration than other practicing physicians.

These findings are consistent with Alraqibah et al. [11] who suggested that older physicians (>45 years) who had more years in practice (>10 years) were associated with higher practice scores (*p* < 0.05). Corroborating these reports, Peng et al. [15] documented that better practice levels were more common among highly educated physicians, those with senior job titles, and those with more years of experience. However, in previous reports by Alqarni and Hassan [14], significant predictors of ULT misuse included the proportion of patients with gout or AH receiving ULT among people visiting patients, rheumatology conference attendance, and having a close relative with AH or gout.

Moreover, we noted a strong positive correlation between knowledge and practice scores, which indicates that their practices will also likely increase when physicians’ knowledge increases. This effect could be due to physicians’ ability to make informed, evidence-based decisions, educate patients effectively, and adapt to new clinical guidelines, all of which enhance patient outcomes. This mirrored the reports of Tawhari et al. [16] indicating that knowledge was strongly correlated with practice (*p* < 0.001). Supporting these reports, Peng et al. [15] documented that higher scores related to attitude were significantly associated with proactive practices (*p* < 0.001).

### 4.6. Management of Patients with Gout

In assessing the data regarding practice, we noted that many physicians were unaware of the most appropriate diagnostic tools for evaluating patients with suspected gouty attacks during the initial evaluation. Their confidence in prescribing anti-inflammatory drugs as an initial treatment for acute gout flare was low; most of them were unaware of the correct duration (3–6 months) for anti-inflammatory prophylaxis used following the initiation of ULT for gout. Furthermore, despite the majority knowing that AH and first gout flare were conditions indicating they should not consider initiating ULT, that lack of recognition contributed to significantly elevated SUA levels (42.7%) and the first gout flare with urolithiasis (32.5%). In addition, physicians do not clearly recognize the typical target range for SUA levels after starting ULT for gout (46.5%), nor do they recognize the duration of ULT use for patients with gout (33.3%).

These findings are consistent with those of Edar et al. [23] who observed that general practitioners exhibited poor practice in managing patients with gout. In contrast, a study by Hidayat et al. [24] observed better management of gout patients. They found that approximately 81% of physicians did not recommend synovial fluid examination to suspected patients with gout, while more than half (52.6%) prescribed allopurinol to AH patients. They also prescribed allopurinol therapy during acute gout incidents and usually discontinued it when SUA normalized (61.8%). The key reasons for these differences include different treatment guidelines, medication preferences, cultural and regional variations, and patient-centered care. A multifaceted approach may lead to improved physician practices in managing gout patients, including education and training, access to evidence-based guidelines, quality improvement initiatives, research and development, awareness, and advocacy.

### 4.7. Recommended Dietary Patterns

We based our dietary recommendations questionnaire on the 2020 American College of Rheumatology guidelines [12]. Despite gaps in the treatment patterns for gout patients, physicians observed improved dietary patterns of gout patients. For instance, most physicians recommend avoiding organ meats high in purine content as well as alcohol overuse; however, only 39.2% recognized high fructose corn syrup-sweetened soda as a dietary pattern that should also be avoided.

However, recommendations to limit nutritional patterns achieved unfavorable ratings, particularly regarding beef or lamb (42.5%), alcohol (34.3%), servings of naturally sweet fruit juices (33.7%), seafood with high purine content (31.5%), and table salt (28.8%). The practice of prescribing physicians in Indonesia is noteworthy; the majority were aware that red meat, seafood, purine-rich vegetables, and alcohol consumption could potentially lead to gouty arthritis, while dairy products may not [24]. In support of this report, Alqarni and Hassan [14] reported that primary healthcare physicians in the Western Region of Saudi Arabia correctly recommended lifestyle changes and a low purine diet for patients with gout.

This study has several limitations. First, the use of a convenience sampling method may limit the generalizability of the findings to all physicians in Saudi Arabia. Second, the self-reported nature of the questionnaire may introduce response bias. Third, the knowledge section included only three items, which may limit the ability to fully assess physicians’ understanding of AH and is considered conceptually thin. Therefore, future studies should increase the number of items used to examine the knowledge in order to enhance the internal consistency and content coverage.

Our study used convenience sampling via social media, which may have led to overrepresentation of younger, digitally engaged physicians. National data on regional and specialty distribution are not publicly available, limiting assessment of representativeness. These factors may introduce selection and nonresponse biases. The knowledge assessment included multiple-choice items with predefined correct answers based on the Al-Arfaj nationwide study, Du, L. et al., and US NHANES [1,2,13]. Participants’ responses could not be re-scored using alternative thresholds, which may limit comparability across guideline definitions. Additionally, a few dietary knowledge items may have variations in interpretation across sources. While answers were verified against current guidelines, minor differences in scoring could exist, which is a limitation of the questionnaire format. Future research should consider larger, randomized samples and include more comprehensive knowledge assessments. Targeted continuing medical education programs focusing on asymptomatic hyperuricemia gout are recommended, especially for younger and less experienced physicians.

## 5. Conclusions

Physicians’ knowledge and practices for managing AH or gout are lacking. Older physicians who were orthopedic consultants and actively read more about AH or gout were likely to demonstrate better knowledge and practice in managing these patients. Furthermore, CME attendance was not a relevant factor in knowledge, and in fact, it seemed to be associated with lower practice score. However, seeking and reading more information about the disease and awareness of the latest guidelines on the management of gout significantly improved both knowledge and practice. Interestingly, evidence suggests a positive link between knowledge and practice.

There is a need to improve physicians’ knowledge and practice in managing AH or gout cases. Improving physicians’ perspectives requires several strategic approaches, including regular CME attendance, access to updated guidelines, interdisciplinary collaboration, technology utilization, research, evidence-based practice participation, patient education, and communication.

## Figures and Tables

**Figure 1 healthcare-13-02719-f001:**
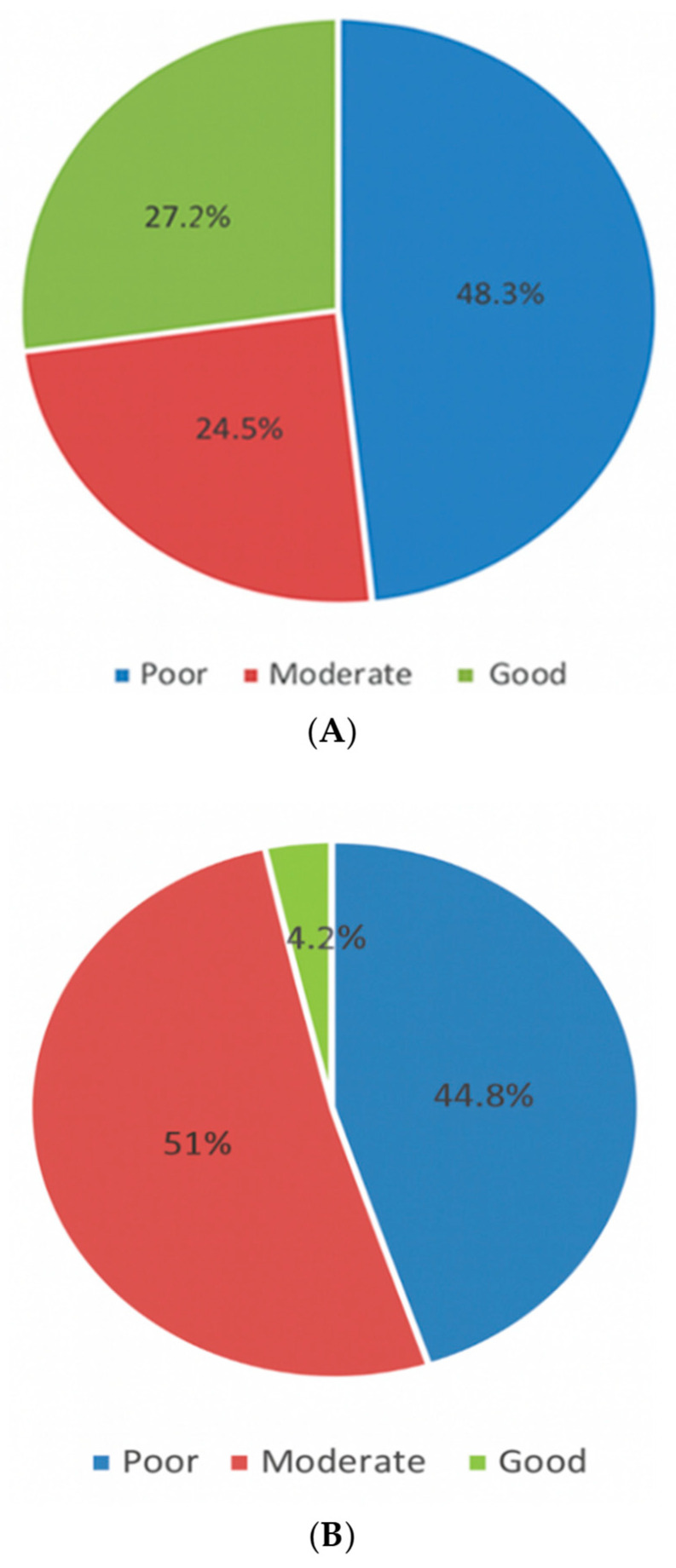
(**A**): Level of knowledge in the management of AH and gout; (**B**): Level of practice in the management of AH and gout.

**Figure 2 healthcare-13-02719-f002:**
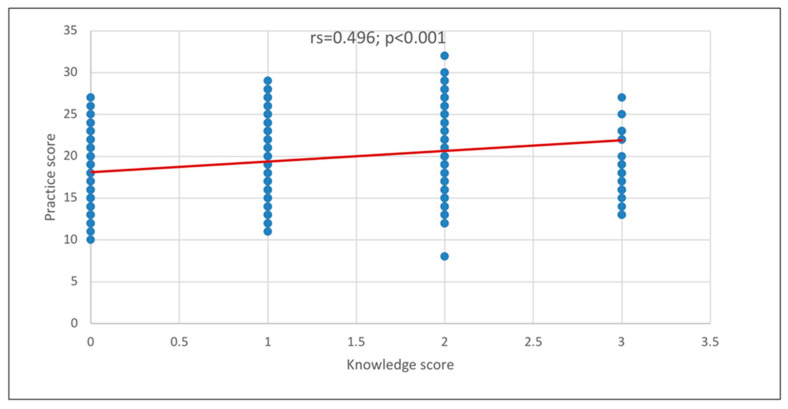
Correlation between knowledge and practice scores.

**Figure 3 healthcare-13-02719-f003:**
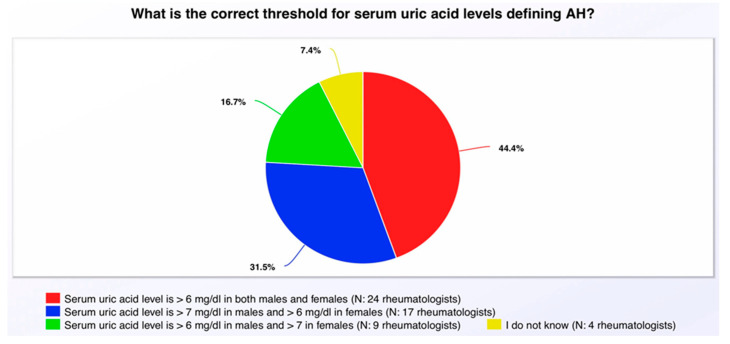
Answers of rheumatologists to the first question in the knowledge section of the questionnaire.

**Table 1 healthcare-13-02719-t001:** Sociodemographic characteristics of physicians (n = 744).

Study Variables	N (%)
Age group	
● 24–34 years	440 (59.1%)
● 35–44 years	164 (22.0%)
● 45–54 years	104 (14.0%)
● 55–64 years	26 (03.5%)
● >64 years	10 (01.3%)
Gender	
● Male	343 (46.1%)
● Female	401 (53.9%)
Region of practice	
● Central Region	172 (31.3%)
● Southern Region	87 (11.7%)
● Western Region	344 (46.2%)
● Eastern Region	104 (14.0%)
● Northern Region	37 (05.0%)
Medical specialty	
● General practitioner	233 (31.3%)
● Rheumatology	54 (07.3%)
● Internal medicine	147 (19.8%)
● Emergency medicine	71 (09.5%)
● Family medicine	184 (24.7%)
● Orthopedic surgery	55 (07.4%)
Current level of medical training/practice	
● Physician without specialty training	311 (41.8%)
● Resident	207 (27.8%)
● Fellow	66 (08.9%)
● Specialist/ Registrar	91 (12.2%)
● Consultant	69 (09.3%)
Years of experience (after internship completion)	
● <5 years	383 (51.5%)
● 5–10 years	240 (32.3%)
● >10 years	121 (16.3%)
In the past three years, have you attended any CME activities specifically focused on AH or gout?	
● Yes	437 (58.7%)
● No	307 (41.3%)
In the past three years, have you actively sought out and reviewed information about AH or gout (e.g., through medical journals, online resources, textbooks)?	
● Yes	497 (66.8%)
● No	247 (33.2%)
Are you aware of the latest 2020 American College of Rheumatology guidelines for the management of gout?	
● Yes	436 (58.6%)
● No	308 (41.4%)

**Table 2 healthcare-13-02719-t002:** Assessment of physicians’ knowledge and practice in the management of AH and gout (n = 744).

Knowledge Items * Indicates the Correct Answer	N (%)
1. What is the correct threshold for serum uric acid levels defining AH? [Correct answer: Serum uric acid level is >7 mg/dL in males and >6 mg/dL in females] *	364 (48.9%)
[Serum uric acid level is >6 mg/dL in both males and females]	168 (22.6%)
[Serum uric acid level is >6 mg/dL in males and >7 in females]	90 (12.1%)
[I do not know]	122 (16.4%)
2. AH always progresses to gouty arthritis [Correct answer: incorrect] *	407(54.7%)
[correct]	204 (27.4%)
[I do not know]	133 (17.9%)
3. AH always needs treatment [Correct answer: incorrect] *	411 (55.2%)
[correct]	225 (30.2%)
[I do not know]	108 (14.6%)
Total knowledge score (mean ± SD)	1.59 ± 1.09
Level of knowledge about AH	
● Poor	359 (48.3%)
● Moderate	182 (24.5%)
● Good	203 (27.3%)
**Practice items** *** Indicates the correct answer**	
Which of the following is MOST appropriate to perform during the initial evaluation of a patient with a suspected gouty attack? [Correct answer: Joint aspiration and analysis of synovial fluid for crystals] *	374 (50.3%)
2. Which approach do you typically take to initiating treatment for a patient with acute gout flare? [Correct answer: Start anti-inflammatory drugs (colchicine, NSAID, or steroid) in combination with urate-lowering therapy at the same time] *	219 (29.4%)
3. Which type of anti-inflammatory medication do you typically prescribe first for managing a gout flare? [Correct answer: I typically consider any of the above drugs depending on the patient’s situation] *	233 (31.3%)
4. For how long can anti-inflammatory prophylaxis (NSAID, colchicine, or steroids) typically be continued following the initiation of urate-lowering therapy (ULT) for gout? [Correct answer: 3–6 months] *	221 (29.7%)
Under which of the following conditions would you typically initiate ULT? (Select all that apply)	
5. AH [Correct answer: No, should leave and not select choice]	569 (76.5%)
6. First gout flare [Correct answer: No, should leave and not select choice]	519 (69.8%)
7. More than one gout flare per year [Correct answer: yes, should select choice and not leave] *	403 (54.2%)
8. First gout flare with any of the following: CKD (chronic kidney disease) of stage 3 or higher, radiographic damage attributable to gout, presence of one or more tophi [Correct answer: Yes, should select choice and not leave] *	390 (52.4%)
9. First gout flare WITH a significantly elevated serum uric acid level (e.g., >9 mg/dL) [Correct answer: yes, should select choice and not leave] *	318 (42.7%)
10. First gout flare WITH urolithiasis [Correct answer: yes, should select choice and not leave] *	242 (32.5%)
11. When initiating allopurinol for ULT in patients with gout, what is the typical starting dose you prescribe in most cases? [Correct answer: 100 mg once daily and titrate up] *	392 (52.7%)
12. What is the typical target range for serum uric acid levels AFTER starting ULT for gout? [Correct answer: <6mg/dL (<360 µmol/L)] *	346 (46.5%)
13. If the target serum uric acid range has not been achieved after 3–6 weeks of an ULT, which of the following is your MOST LIKELY course of action? [Correct answer: Increase the dose of the same ULT or switch to another ULT] *	446 (59.9%)
14. In most cases, how long should ULT be continued for patients with gout? [Correct answer: Indefinite with regular follow-up uric acid level]*	248 (33.3%)
15. How do you typically address diet and lifestyle modifications for gout or hyperuricemia management in your clinical practice? [Optimal answer: I routinely discuss the importance of diet and lifestyle modifications with patients and provide resources or referrals] *	368 (49.5%)
16. * *	
[I acknowledge the importance of diet and lifestyle modifications but have limited time for in-depth discussions. I may provide basic guidance or handouts]	303 (40.7%)
[Diet and lifestyle modifications are not routinely addressed during consultations]	73 (9.8%)
Which of the following diet items would you recommend a patient with gout to AVOID? (Select all that apply)	
17. Organ meats high in purine content (e.g., liver, kidney) [Correct answer: yes, should select choice and not leave] *	437 (58.7%)
18. High fructose corn syrup-sweetened soda [Correct answer: yes, should select choice and not leave] *	310 (41.7%)
19. Beef or lamb [Correct answer: No, should leave and not select choice]	398 (53.5%)
20. Seafood with high purine content (e.g., sardines, shellfish) [Correct answer: No, should leave and not select choice]	380 (51.1%)
21. Alcohol overuse (defined as more than 2 servings per day for a male and 1 serving per day for a female) [Correct answer: yes, should select choice and not leave] *	367 (49.3%)
22. Alcohol (small amount) [Correct answer: No, should leave and not select choice]	377 (50.7%)
23. Servings of naturally sweet fruit juices (high fructose juices such as grape and apple juice) [Correct answer: No, should leave and not select choice]	537 (72.2%)
24. Table sugar, sweetened beverages, and desserts [Correct answer: No, should leave and not select choice]	537 (72.2%)
25. Table salt, including sauces [Correct answer: No, should leave and not select choice]	603 (81.0%)
26. Low-fat or non-fat dairy [Correct answer: No, should leave and not select choice]	680 (91.4%)
27. Vegetable [Correct answer: No, should leave and not select choice] 28. I do not know	675 (90.7%) 45 (6.04%)
Which of the following diet items would you recommend a patient with gout to LIMIT? (Select all that apply)	
29. Organ meats high in purine content (e.g., liver, kidney) [Correct answer: No, should leave and not select choice]	455 (61.2%)
30. High fructose corn syrup-sweetened soda [Correct answer: No, should leave and not select choice]	523 (70.3%)
31. Beef or lamb [Correct answer: yes, should select choice and not leave] *	316 (42.5%)
32. Seafood with high purine content (e.g., sardines, shellfish) [Correct answer: yes, should select choice and not leave] *	234 (31.5%)
33. Alcohol overuse (defined as more than two servings per day for a male and one serving per day for a female) [Correct answer: No, should leave and not select choice]	489 (65.7%)
34. Alcohol (small amount) [Correct answer: yes, should select choice and not leave] *	255 (34.3%)
35. Servings of naturally sweet fruit juices (high fructose juices such as grape and apple juice) [Correct answer: yes, should select choice and not leave] *	251 (33.7%)
36. Table sugar, sweetened beverages, and desserts [Correct answer: yes, should select choice and not leave] *	251 (33.7%)
37. Table salt, including sauces [Correct answer: yes, should select choice and not leave] *	214 (28.8%)
38. Low-fat or non-fat dairy [Correct answer: No, should leave and not select choice]	607 (81.6%)
39. Vegetable [Correct answer: No, should leave and not select choice]	619 (83.2%)
40. I do not know	63 (8.4%)
Total practice score (mean ± SD)	19.9 ± 4.16
Level of practice	
● Poor	333 (44.8%)
● Moderate	380 (51.0%)
● Good	31 (4.2%)

* Indicates the correct answer.

**Table 3 healthcare-13-02719-t003:** Association between knowledge and the sociodemographic characteristics of the physicians (n = 744).

Factor	Knowledge Score (3) Mean ± SD	Z/H-Test	*p*-Value
Age group ^a^			
● <35 years	1.44 ± 1.07	4.518	<0.001 **
● ≥35 years	1.81 ± 1.09		
Gender ^a^			
● Male	1.73 ± 1.08	3.380	0.001 **
● Female	1.46 ± 1.09		
Region of practice ^a^			
● Inside Western Region	1.72 ± 1.05	2.964	0.003 **
● Outside Western Region	1.48 ± 1.12		
Medical specialty ^b^			
● General practitioner	1.33 ± 1.09	34.403	<0.001 **
● Rheumatology	1.35 ± 0.91		
● Internal medicine	1.72 ± 1.09		
● Emergency medicine	1.46 ± 1.07		
● Family medicine	1.78 ± 1.05		
● Orthopedic surgery	2.05 ± 1.11		
Current level of medical training/practice ^b^			
● Physician without specialty training	1.42 ± 1.06	31.791	<0.001 **
● Resident	1.65 ± 1.10		
● Fellow	1.35 ± 0.97		
● Specialist/Registrar	1.81 ± 1.13		
● Consultant	2.12 ± 1.02		
Years of experience (after internship completion) ^a^			
● <5 years	1.42 ± 1.07	4.417	<0.001 **
● ≥5 years	1.77 ± 1.09		
In the past three years, have you attended any CME activities specifically focused on AH or gout? ^a^			
● Yes	1.53 ± 1.08	1.765	0.078
● No	1.67 ± 1.09		
In the three years, have you actively sought out and reviewed information about AH or gout? ^a^			
● Yes	1.70 ± 1.09	3.997	<0.001 **
● No	1.36 ± 1.07		
Are you aware of the latest 2020 American College of Rheumatology guidelines for the management of gout? ^a^			
● Yes	1.63 ± 1.07	1.065	0.287
● No	1.54 ± 1.12		

^a^ *p*-value has been calculated using the Mann–Whitney Z-test. ^b^ *p*-value has been calculated using the Kruskal–Wallis H-test. ** Significant at *p* < 0.05 level.

**Table 4 healthcare-13-02719-t004:** Association between practice and the sociodemographic characteristics of the physicians (n = 744).

Factor	Practice Score (37) Mean ± SD	Z/H-Test	*p*-Value
Age group ^a^			
● <35 years	19.2 ± 3.91	5.643	<0.001 **
● ≥35 years	20.9 ± 4.29		
Gender ^a^			
● Male	20.4 ± 4.38	2.882	0.004 **
● Female	19.5 ± 3.92		
Region of practice ^a^			
● Inside Western Region	20.3 ± 4.29	2.581	0.010 **
● Outside Western Region	19.5 ± 4.02		
Medical specialty ^b^			
● General practitioner	18.7 ± 3.50	26.529	<0.001 **
● Rheumatology	19.6 ± 3.97		
● Internal medicine	20.4 ± 4.18		
● Emergency medicine	20.0 ± 4.35		
● Family medicine	20.6 ± 4.52		
● Orthopedic surgery	21.3 ± 4.38		
Current level of medical training/practice ^b^			
● Physician without specialty training	18.9 ± 3.92	45.425	<0.001 **
● Resident	20.4 ± 4.32		
● Fellow	19.2 ± 3.61		
● Specialist/Registrar	20.7 ± 3.96		
● Consultant	22.2 ± 4.24		
Years of experience (after internship completion) ^a^			
● <5 years	19.3 ± 4.03	4.172	<0.001 **
● ≥5 years	20.6 ± 4.21		
In the past three years, have you attended any CME activities specifically focused on AH or gout? ^a^			
● Yes	19.5 ± 4.23	3.353	0.001 **
● No	20.4 ± 4.02		
In the three years, have you actively sought out and reviewed information about AH or gout? ^a^			
● Yes	20.3 ± 4.19	3.267	0.001 **
● No	19.1 ± 3.99		
Are you aware of the latest 2020 American College of Rheumatology guidelines for the management of gout? ^a^			
● Yes	20.3 ± 4.48	2.017	0.044 **
● No	19.4 ± 3.61		

^a^ *p*-value was calculated using the Mann–Whitney Z-test. ^b^ *p*-value has been calculated using the Kruskal–Wallis H-test. ** Significant at *p* < 0.05 level.

**Table 5 healthcare-13-02719-t005:** Multiple mean differences of knowledge scores in relation to the level of medical practice (n = 744).

(I) Level of Medical Training/Practice	(J) Level of Medical Training/Practice	Mean Difference (I–J)	Std. Error	Sig.	95% Confidence Interval	
					Lower Bound	Upper Bound
Physician without specialty training	Specialist/Registrar	−0.39518 *	0.12773	0.021	−0.7548	−0.0356
	Consultant	−0.69794 *	0.14262	0.000	−1.0995	−0.2964
Resident	Consultant	−0.46860 *	0.14898	0.017	−0.8880	−0.0492
Fellow	Consultant	−0.76746 *	0.18452	0.000	−1.2870	−0.2479

* The mean difference is significant at 0.05.

**Table 6 healthcare-13-02719-t006:** Multiple mean differences of practice scores in relation to the level of medical practice (n = 744).

(I) Level of Medical Training/Practice	(J) Level of Medical Training/Practice	Mean Difference (I–J)	Std. Error	Sig.	95% Confidence Interval	
					Lower Bound	Upper Bound
Physician without specialty training	Resident	−1.43632 *	0.36271	0.001	−2.4575	−0.4151
	Specialist/Registrar	−1.73732 *	0.48191	0.003	−3.0941	−0.3805
	Consultant	−3.27690 *	0.53807	0.000	−4.7918	−1.7620
Resident	Consultant	−1.84058 *	0.56208	0.011	−3.4231	−0.2580
Fellow	Consultant	−2.98946 *	0.69619	0.000	−4.9496	−1.0294

* The mean difference is significant at the 0.05 level.

**Table 7 healthcare-13-02719-t007:** Knowledge and practice scores across specialties.

Specialty	N (%)	Knowledge Score Mean ± SD	Practice Score Mean ± SD
Orthopedic			
● Physician without specialty training	07 (12.7%)	1.86 ± 1.21	22.6 ± 4.69
● Resident	18 (32.7%)	2.56 ± 0.70	23.2 ± 4.09
● Fellow	03 (05.5%)	1.00 ± 1.00	15.3 ± 4.04
● Specialist/Registrar	17 (30.9%)	1.71 ± 1.21	19.0 ± 2.35
● Consultant	10 (18.2%)	2.20 ± 1.23	22.4 ± 4.84
Family medicine			
● Physician without specialty training	35 (19.0%)	1.94 ± 0.91	19.6 ± 4.96
● Resident	85 (46.2%)	1.80 ± 1.06	20.8 ± 4.55
● Fellow	17 (09.2%)	1.35 ± 0.99	19.3 ± 4.24
● Specialist/Registrar	28 (15.2%)	1.61 ± 1.19	21.2 ± 4.17
● Consultant	19 (10.3%)	2.05 ± 0.97	21.6 ± 4.09
Internal medicine			
● Physician without specialty training	56 (38.1%)	1.23 ± 1.06	19.1 ± 4.09
● Resident	31 (21.1%)	1.71 ± 1.07	19.7 ± 4.36
● Fellow	16 (10.9%)	1.39 ± 0.95	20.4 ± 3.46
● Specialist/Registrar	23 (15.6%)	2.39 ± 0.84	22.4 ± 3.57
● Consultant	21 (14.3%)	2.33 ± 0.97	22.7 ± 3.79
General practitioner	233 (100%)	1.33 ± 1.09	18.7 ± 3.50
Rheumatology			
● Physician without specialty training	11 (20.4%)	1.00 ± 0.77	16.8 ± 2.56
● Resident	11 (20.4%)	1.00 ± 1.00	19.1 ± 3.24
● Fellow	15 (27.8%)	1.00 ± 0.65	19.6 ± 2.75
● Specialist/Registrar	07 (13.0%)	1.43 ± 0.79	18.0 ± 4.73
● Consultant	10 (18.5%)	1.60 ± 0.52	24.6 ± 2.79
Emergency medicine			
● Physician without specialty training	21 (29.6%)	1.05 ± 0.74	19.5 ± 4.52
● Resident	30 (42.3%)	1.20 ± 0.85	20.7 ± 4.51
● Fellow	08 (11.3%)	0.88 ± 0.64	18.6 ± 4.10
● Specialist/Registrar	07 (09.9%)	1.43 ± 0.79	20.1 ± 3.53
● Consultant	05 (07.0%)	1.40 ± 0.89	20.4 ± 5.03

**Table 8 healthcare-13-02719-t008:** Multiple linear regression analysis to determine the effect of CME attendance in relation to practice (n = 744).

Factor	β	95% CI	*p*-Value
In the past three years, have you attended any CME activities specifically focused on AH or gout? ^a^	1.368	0.738–1.998	<0.001 **

^a^: Adjusted for age, gender, years in practice, speciality role, and guidelines awareness. ** Significant at *p* < 0.05 level.

## Data Availability

The original contributions presented in this study are included in the article. Further inquiries can be directed to the corresponding author.

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
