# Peer review of "Knowledge and Treatment of Asymptomatic Hyperuricemia Versus Gout Among Physicians in Saudi Arabia: A Cross-Sectional Survey"

_healthcare, 2025, doi:10.3390/healthcare13212719_

Round 1

Reviewer 1 Report

Comments and Suggestions for Authors

I appreciate the authors for conducting this important nationwide survey on physicians’ knowledge and treatment practices regarding asymptomatic hyperuricemia versus gout in Saudi Arabia

Comments on Abstract

  • Line 13: Replace the heading “Introduction” with “Background”.
  • Line 15: Delete the standalone heading “Aim”. Instead, integrate the aim into the Background section text.

Line no 17 instead o f subject and methods write only methods

Line no 22 and 23 font size is very small make it uniform

Authors can modify the methods based on the given below Instead of writing inclusion and exclusion in details in abstract.

Methods: A standardized, online validated questionnaire was used to collect data from physicians in Saudi Arabia. The tool comprised two parts: a 3-item section on demographics and a 37-item section assessing knowledge (please mention the number of items) and practice (please mention the number of items) related to the management of asymptomatic hyperuricemia (AH) and gout. Informed consent was obtained from all participants. Please specify the type of sampling procedure used. Clarify the statistical methods applied. State that p < 0.05 was considered statistically significant.

Did authors do the validation of the questionnaire??

Provide details?

Results

Only descriptive is given.. no statistical values given….

Introduction

 The current introduction provides a basic overview of AH and gout but is too brief. Expand it with recent studies from the past 5–10 years, both globally and in Saudi Arabia. Include prevalence, risk factors, and clinical significance.

Importance of physician knowledge and rationale for the study.

Include a clear knowledge gap to justify the study.

Incorporate recent studies with more informative content

The hypothesis can be stated before the aim to clarify the study’s focus and expectations.Materials and methods

Comments on Methods Section

  1. Study Design and Setting
    • Please mention the duration of the study (start and end months/years).
  2. Section Order
    • Move Ethical Considerations to follow Statistical Analysis for better logical flow.

2.3. Sampling Technique and Participants

Mention the tool or formula used to calculate the sample size.

Provide detailed inclusion and exclusion criteria here (these can be adapted from the Abstract Methods section).

2.4. Study Measurements

State that the questionnaire’s content validity was confirmed.

Provide details of the pilot study:

Number of participants included.

Whether any items, wording, or structure were modified based on pilot feedback.

Explain how validity was established (e.g., expert panel review, content validity index).

Provide the reliability statistics

2.5. Questionnaire Criteria

Please specify the number of questions included in the knowledge section and the practice section.

Also describe questions in each section

Describe the type of response format used (e.g., Yes/No, multiple-choice, Likert scale).

The following statement is already mentioned in the previous section and should be deleted here to avoid repetition: “The questionnaire was validated, and a pilot study confirmed its reliability.”

2.7. Definition of general practitioner and general physician

Move this section to before 2.4. Study Measurements.

Discussion

  1. Well written
  2. Provide Limitations and recommendations

Major revision is needed

Author Response

Dear Reviewer 1,

  • Line 13: Replace the heading “Introduction” with “Background”.
  • Line 15: Delete the standalone heading “Aim”. Instead, integrate the aim into the Background section text.

Line no 17 instead o f subject and methods write only methods

Line no 22 and 23 font size is very small make it uniform

Authors can modify the methods based on the given below Instead of writing inclusion and exclusion in details in abstract.

Methods: A standardized, online validated questionnaire was used to collect data from physicians in Saudi Arabia. The tool comprised two parts: a 3-item section on demographics and a 37-item section assessing knowledge (please mention the number of items) and practice (please mention the number of items) related to the management of asymptomatic hyperuricemia (AH) and gout. Informed consent was obtained from all participants. Please specify the type of sampling procedure used. Clarify the statistical methods applied. State that p < 0.05 was considered statistically significant.

Did authors do the validation of the questionnaire??

Provide details?

Results 

Only descriptive is given.. no statistical values given….

Response: Thank you for your valuable comments. All your comments were done as you addressed, see page 1.

Introduction

 The current introduction provides a basic overview of AH and gout but is too brief. Expand it with recent studies from the past 5–10 years, both globally and in Saudi Arabia. Include prevalence, risk factors, and clinical significance.

Importance of physician knowledge and rationale for the study.

Include a clear knowledge gap to justify the study.

Incorporate recent studies with more informative content

The hypothesis can be stated before the aim to clarify the study’s focus and expectations.Materials and methods

Response: Thank you for the constructive suggestions. All the suggested modifications have been completed in the introduction, see page 2.

Comments on Methods Section

  1. Study Design and Setting
    • Please mention the duration of the study (start and end months/years).
  2. Section Order
    • Move Ethical Considerations to follow Statistical Analysis for better logical flow.

2.3. Sampling Technique and Participants

Mention the tool or formula used to calculate the sample size.

Provide detailed inclusion and exclusion criteria here (these can be adapted from the Abstract Methods section).

2.4. Study Measurements

State that the questionnaire’s content validity was confirmed.

Provide details of the pilot study:

Number of participants included.

Whether any items, wording, or structure were modified based on pilot feedback.

Explain how validity was established (e.g., expert panel review, content validity index).

Provide the reliability statistics

2.5. Questionnaire Criteria

Please specify the number of questions included in the knowledge section and the practice section.

Also describe questions in each section

Describe the type of response format used (e.g., Yes/No, multiple-choice, Likert scale).

The following statement is already mentioned in the previous section and should be deleted here to avoid repetition: “The questionnaire was validated, and a pilot study confirmed its reliability.”

2.7. Definition of general practitioner and general physician 

Move this section to before 2.4. Study Measurements.

Response: All suggested modifications have been completed, see pages 3 and 4.

Discussion 

  1. Well written
  2. Provide Limitations and recommendations

Response: Thank you for your suggestion. A limitation and recommendation paragraph was added, see page 22.  

Reviewer 2 Report

Comments and Suggestions for Authors

This study is a national survey evaluating physicians' knowledge and treatment practices regarding asymptomatic hyperuricemia (AH) and gout in Saudi Arabia. The study, which included 744 physicians, reveals that participants generally have inadequate knowledge and practice in managing AD and gout, particularly among younger individuals, women, and general practitioners. The article contains several significant shortcomings:

Lines 98-114: The categories (poor/fair/good) used for knowledge (3 questions) and practice (37 questions) scores appear reasonable. However, whether a scale with only 3 questions can reliably measure knowledge is debatable. The lack of an internal consistency coefficient, such as Cronbach's Alpha, in the pilot study results creates a lack of reliability in the measurement tool.

Lines 119-125: In the statistical analysis, the normality test indicated that the scores were normally distributed, but non-parametric tests (Mann-Whitney U, Kruskal-Wallis) were used instead of parametric tests. This is a contradiction and suggests an error in the statistical method selection.

Line 144: The title of Table 1 states n-74, but the text and table content indicate 744 participants. Could this be a misrepresentation?

Table 3 only compares the Inside Western Region and Outside Western Region for the Region of Practice variable. However, Table 1 presents five distinct regions (Central, Southern, Western, Eastern, and Northern). It is not explained why the analysis was conducted as a binary grouping or why differences between other regions were not examined.

Lines 419-449: The conclusion states that CME seems to increase only in physicians' practice. However, Table 3 shows no significant difference between CME participation and knowledge scores (p=0.078), while Table 4 shows that those who did not participate in CME had higher practice scores (p=0.001). These findings do not fully align with the conclusion and indicate confusion in interpreting the findings.

Tables 5 and 6, which present the post-hoc analysis results, present a confusing list of groups where significant differences occurred. The format of the tables hinders readability. A more straightforward presentation (listing groups one below the other and highlighting only significant differences) would have been preferable.

While this study provides a valuable national dataset that highlights significant knowledge and practice gaps in the management of AD and gout among physicians in Saudi Arabia, it also has some methodological limitations that require attention, particularly regarding the reliability of the measurement tools, the choice of statistical method, and the interpretation of the findings. Therefore, this study deserves a second evaluation. However, this article requires a major revision.

Author Response

Dear Reviewer 2,

Comment: This study is a national survey evaluating physicians' knowledge and treatment practices regarding asymptomatic hyperuricemia (AH) and gout in Saudi Arabia. The study, which included 744 physicians, reveals that participants generally have inadequate knowledge and practice in managing AD and gout, particularly among younger individuals, women, and general practitioners. The article contains several significant shortcomings:

“Lines 98-114: The categories (poor/fair/good) used for knowledge (3 questions) and practice (37 questions) scores appear reasonable. However, whether a scale with only 3 questions can reliably measure knowledge is debatable. The lack of an internal consistency coefficient, such as Cronbach's Alpha, in the pilot study results creates a lack of reliability in the measurement tool.”

Response: We appreciate the reviewer’s comment. However, we would like to clarify that the knowledge and practice items in our questionnaire are not suitable for assessment using Cronbach’s Alpha, as this coefficient is primarily intended to evaluate the internal consistency of scales composed of Likert-type items or items with continuous or ordinal responses. In contrast, our questionnaire consists of dichotomous (e.g., correct/incorrect) and categorical items, which are not appropriate for Cronbach’s Alpha. Alternative reliability measures, such as the Kuder–Richardson Formula 20 (KR-20) was applied, see lines 127-131.

“Lines 119-125: In the statistical analysis, the normality test indicated that the scores were normally distributed, but non-parametric tests (Mann-Whitney U, Kruskal-Wallis) were used instead of parametric tests. This is a contradiction and suggests an error in the statistical method selection.”

Response: The mention of normally distributed data and the application of parametric tests was a typographical error introduced during the final proofreading stage. There are no contradictions in the statistical approach used, as non-parametric tests were consistently applied throughout the analysis. Both the knowledge and practice scores were confirmed to be non-normally distributed based on the Kolmogorov–Smirnov test and visual inspection of distribution plots. Therefore, no changes to the statistical analyses are required. We revised the manuscript accordingly to correct the wording and accurately reflect the use of non-parametric methods based on skewness and kurtosis measures, see lines 170-178.

“Line 144: The title of Table 1 states n-74, but the text and table content indicate 744 participants. Could this be a misrepresentation?”

Response: We thank you for drawing attention to this point. In our original file, the title of Table 1 correctly indicated n = 744 participants, consistent with both the text and table content. It is possible that this discrepancy may have occurred during the manuscript formatting or conversion process. We will carefully review the submitted version to ensure that the correct sample size (n = 744) is consistently presented throughout the manuscript and tables.

“Table 3 only compares the Inside Western Region and Outside Western Region for the Region of Practice variable. However, Table 1 presents five distinct regions (Central, Southern, Western, Eastern, and Northern). It is not explained why the analysis was conducted as a binary grouping or why differences between other regions were not examined.”

Response: Our initial plan was to compare all five regions individually, and we did conduct preliminary analyses in this regard. However, we decided not to present them in the current version because such comparisons require larger subgroup sizes to yield more robust conclusions. To simplify the analysis and maintain statistical validity, we therefore presented the results as a binary comparison (Inside Western Region vs. Outside Western Region) as supplementary information. If the reviewer and editor consider it more appropriate, we are willing to provide the full regional comparisons in the revised manuscript.

“Lines 419-449: The conclusion states that CME seems to increase only in physicians' practice. However, Table 3 shows no significant difference between CME participation and knowledge scores (p=0.078), while Table 4 shows that those who did not participate in CME had higher practice scores (p=0.001). These findings do not fully align with the conclusion and indicate confusion in interpreting the findings.”

Response: We thank you for this insightful observation. Although CME attendance did not emerge as a significant factor in knowledge (p = 0.078, Table 3), an unexpected finding was observed in relation to practice scores. The bivariate analysis suggested that physicians who did not attend CME activities had higher practice scores compared to those who did (p = 0.001, Table 4). To further explore this relationship, we conducted a multiple linear regression model adjusting for potential confounding variables—including age, gender, years in practice, specialty, role, and guideline awareness. The results indicated that CME attendance was positively and independently associated with higher practice scores (β = 1.368; 95% CI = 0.738–1.998; p < 0.001) (Table 8). This suggests that, although the unadjusted results appeared contradictory, the adjusted analysis supports a beneficial effect of CME attendance on physicians’ practice.

Tables 5 and 6, which present the post-hoc analysis results, present a confusing list of groups where significant differences occurred. The format of the tables hinders readability. A more straightforward presentation (listing groups one below the other and highlighting only significant differences) would have been preferable.

Response: We appreciate the reviewer’s feedback. Table 5 and 6 have been revised to present only the significant post-hoc comparisons in a clear and more readable format.

We thank the reviewer for this valuable comment regarding the clarity of Tables 5 and 6. We acknowledge that the current format may hinder readability and make it difficult to identify the significant group differences. In response, we revised the presentation of these tables by listing the comparison groups in a more straightforward format and clearly highlighting only the statistically significant differences. This adjustment enhanced readability and allow readers to more easily interpret the post-hoc analysis results.

While this study provides a valuable national dataset that highlights significant knowledge and practice gaps in the management of AD and gout among physicians in Saudi Arabia, it also has some methodological limitations that require attention, particularly regarding the reliability of the measurement tools, the choice of statistical method, and the interpretation of the findings. Therefore, this study deserves a second evaluation. However, this article requires a major revision.

Response: Thank you for your valuable comments, we have now addressed all comments raised by the reviewer.

Reviewer 3 Report

Comments and Suggestions for Authors

The topic of this nationwide cross-sectional survey is clinically relevant and the sample size is large. However, there are important methodological and statistical issues that limit interpretability and require substantial revision prior to consideration:

1) A first concern is the questionnaire development, validity, and scoring. The knowledge scale includes only three items scored 0/1, for a possible range of 0–3. Such a short instrument is highly vulnerable to measurement error, ceiling or floor effects, and poor discrimination. The authors should justify why three items are sufficient to represent knowledge and provide evidence of content validity, such as expert panel composition, item blueprint, and item-level content validity index. They should also describe pilot procedures and report reliability indices, such as KR-20 or test–retest, for both knowledge and practice sections. Clarification is also needed on how “correct” answers for practice and diet items were keyed. Several keyed responses appear inconsistent with major guidelines. For example, seafood with high purine content is marked as both “not to avoid” and “to limit” in different places. A table mapping each item to the specific guideline statement from which the key was derived is necessary, since as currently written the scoring of the “avoid/limit” sections contains internal contradictions. The practice level cut-points (<50% poor, 50–75% moderate, >75% good) also appear arbitrary. These thresholds should be justified a priori or sensitivity analyses should be performed with alternative cutoffs, item response theory, or weighted scoring, reporting how conclusions change.

2) A second concern relates to statistical analysis inconsistencies. The methods state that a Kolmogorov–Smirnov test was performed and that based on the plot scores follow a normal distribution, and therefore parametric tests were applied. However, the analysis uses Mann–Whitney, Kruskal–Wallis, and Spearman correlations throughout, which are non-parametric. This is contradictory. The authors should report the actual K–S statistics and p-values, state clearly whether distributions were normal, and then either use parametric tests (t-test, ANOVA, Pearson correlation) or remain with non-parametric tests and remove the claim of normality. If they keep non-parametric tests, the post-hoc procedures should be justified accordingly. In addition, multiple subgroup comparisons by age, sex, region, specialty, role, experience, CME, and guideline awareness risk inflating Type I error. The authors should clarify their family-wise error control strategy, for example through Holm or Bonferroni correction, beyond Dunn–Bonferroni, which seems to have been used only for select post-hoc tests. It should be indicated which results remain significant after correction. The finding that not attending CME is associated with higher practice scores (p = .001) should be explored for confounding factors such as seniority or specialty mix. A multivariable analysis, for example a linear regression of practice score adjusting for age, gender, years in practice, specialty, role, and guideline awareness, would allow testing whether the association remains independent.

3) A third set of issues involves sampling, external validity, and representativeness. Recruitment via social media with convenience sampling is pragmatic but may bias toward younger, digitally engaged clinicians, which is reflected in the fact that 59 percent of respondents were aged 24–34. The authors should discuss selection bias and coverage error explicitly, and temper the claim of being a nationwide survey accordingly. If possible, they should compare the regional and specialty distribution of their sample to national physician statistics. The computation of a minimum sample size based on the total physician population is appropriate in theory, but the actual recruitment strategy did not use probability sampling. The sampling frame, number of invitations sent, view/click/participation rates, and eligibility screening should be clarified, so that readers can better judge nonresponse bias.

4) The definition of “general practitioner” versus “general physician” is another source of confusion. The manuscript defines a general practitioner as post-internship without training and a general physician as working in a specialized field without training. This classification may be unusual internationally and risks confusion. The categories should be aligned with standard nomenclature or more clearly justified with examples. Alternatively, the categories could be merged or renamed.

5) Another issue is the alignment of item keying with guidelines. The Introduction and Methods cite the ACR 2020 guideline, but the knowledge item on AH thresholds appears keyed as >7 mg/dL for men and >6 mg/dL for women, while the manuscript later acknowledges varying definitions across guidelines, such as 6.8 mg/dL for both sexes or 6 mg/dL suggested in other sources. The authors should state explicitly which definition they adopted a priori for item keying, acknowledge heterogeneity across guidelines, and consider a sensitivity analysis re-scoring the knowledge item using alternative accepted thresholds, reporting how knowledge categories change. For dietary items, several “correct answers” appear mis-keyed. These should be verified and corrected, and all affected analyses re-run with updated tables, figures, and text. A supplement should be provided with the full questionnaire and answer key.

6) Interpretation and causal language also need correction. The current text sometimes implies causal effects, such as CME “increases practice” or “knowledge leads to improved practice,” which are not warranted in a cross-sectional design. The language should be reframed in terms of associations. Multivariable and interaction analyses could provide a richer understanding of observed patterns, such as specialty by seniority.

7) The abstract and results should report effect sizes, such as mean differences with confidence intervals, alongside p-values for key comparisons, and avoid over-emphasizing percentage labels (“poor,” “moderate,” “good”) without justification for cutoffs. The methods should cite the source questionnaire from Qassim more fully and describe which items were retained, modified, or added, and why. The pilot sample size, criteria for item revision, and reliability coefficients should be included. Typographical errors such as “Ne York” should be corrected, assumption checks added, and the analysis strategy made consistent. In results tables, 95 percent confidence intervals should be added, reference groups clearly labeled, and the direction of differences indicated. After multiple testing correction, only those comparisons that remain significant should be marked as such. The dietary items in particular should be re-tabulated after re-keying, since as currently written the narrative risks inadvertently endorsing misaligned dietary advice. The discussion should expand the limitations, including non-probability sampling, lack of response rate, potential misclassification from brief instruments, possible differential item functioning across specialties, and heterogeneity of AH definitions. Comparisons with prior studies should add contextual explanations such as differences in policy environment, CME availability, or specialty mix, and avoid selective citation.

8) Regarding ethics and transparency, IRB approval and consent are reported, which is positive, but more detail should be added about data confidentiality procedures, for example whether IP addresses were handled or duplicate submissions prevented, which is standard in online surveys. The data availability statement that “data are included in the article” is insufficient. The anonymized dataset and the full questionnaire with the key should be deposited in a public repository such as Zenodo or OSF and the accession link provided. Funding and conflicts of interest statements appear clear.

9) The references cited are relevant but the background could be better updated and balanced. Recent high-quality guidelines on AH thresholds, treat-to-target serum urate, initiation of ULT during flares, and dietary recommendations should be included, and the item keying aligned accordingly. Some references are dated or regional; including more recent consensus sources would strengthen the foundation. Presentation and clarity can also be improved by fixing typographical and formatting errors, ensuring all tables and figures are correctly numbered and cross-referenced, and providing supplements with the full instrument, scoring guide, pilot details, re-keyed diet items with citations, and a statistical analysis plan that matches the executed analyses.

I look forward with interest to receiving the revised version of the manuscript and to seeing how the authors address these important points.

Comments on the Quality of English Language

The English is understandable, but the quality can be improved to ensure clarity and precision. Several sentences are long and complex, which makes the text harder to follow. Grammar and syntax should be revised for smoother flow, and some technical terms and guideline references should be expressed more consistently. A careful language edit by a native or professional editor would strengthen readability.

Author Response

Dear Reviewer 3,

Reply for reviewer 3:

“1) A first concern is the questionnaire development, validity, and scoring. The knowledge scale includes only three items scored 0/1, for a possible range of 0–3. Such a short instrument is highly vulnerable to measurement error, ceiling or floor effects, and poor discrimination. The authors should justify why three items are sufficient to represent knowledge and provide evidence of content validity, such as expert panel composition, item blueprint, and item-level content validity index. They should also describe pilot procedures and report reliability indices, such as KR-20 or test–retest, for both knowledge and practice sections.”

Response: We sincerely thank the reviewer for raising this important point. We would like to clarify that the main objective of our study was to assess physicians’ clinical practice (37 items), and the knowledge section was included only as a supportive component to provide context. The original validated questionnaire contained 5 knowledge items, but two were excluded as they were not directly relevant to our research focus on clinical practice. We retained three knowledge items that were judged to be most clinically meaningful for our purpose. Additionally, we reported in the limitation section that 3 items may not assess the knowledge efficiently, and we added the findings of KR-20 measure in the method section to assess the reliability, see lines 127-132 and lines 523-525.

“Clarification is also needed on how “correct” answers for practice and diet items were keyed. Several keyed responses appear inconsistent with major guidelines. For example, seafood with high purine content is marked as both “not to avoid” and “to limit” in different places. A table mapping each item to the specific guideline statement from which the key was derived is necessary, since as currently written the scoring of the “avoid/limit” sections contains internal contradictions.”

Response: We have now addressed the reviewer comment in the method section, see lines 156-163 and supplementary table 1.

“The practice level cut-points (<50% poor, 50–75% moderate, >75% good) also appear arbitrary. These thresholds should be justified a priori or sensitivity analyses should be performed with alternative cutoffs, item response theory, or weighted scoring, reporting how conclusions change.”

Response: We than the reviewer for this suggestion. In response. We conducted a sensitivity analysis using quartile-based cutoffs (≤17 = poor, 18–22 = moderate, ≥23 = good), and the overall conclusions remained unchanged and the additional analysis has been added to the manuscript. 

“2) A second concern relates to statistical analysis inconsistencies. The methods state that a Kolmogorov–Smirnov test was performed and that based on the plot scores follow a normal distribution, and therefore parametric tests were applied. However, the analysis uses Mann–Whitney, Kruskal–Wallis, and Spearman correlations throughout, which are non-parametric. This is contradictory. The authors should report the actual K–S statistics and p-values, state clearly whether distributions were normal, and then either use parametric tests (t-test, ANOVA, Pearson correlation) or remain with non-parametric tests and remove the claim of normality. If they keep non-parametric tests, the post-hoc procedures should be justified accordingly. In addition, multiple subgroup comparisons by age, sex, region, specialty, role, experience, CME, and guideline awareness risk inflating Type I error. The authors should clarify their family-wise error control strategy, for example through Holm or Bonferroni correction, beyond Dunn–Bonferroni, which seems to have been used only for select post-hoc tests. It should be indicated which results remain significant after correction.”

Response: The mention of normally distributed data and the application of parametric tests was a typographical error introduced during the final proofreading stage. There are no contradictions in the statistical approach used, as non-parametric tests were consistently applied throughout the analysis. Both the knowledge and practice scores were confirmed to be non-normally distributed based on the Kolmogorov–Smirnov test and visual inspection of distribution plots. Therefore, no changes to the statistical analyses are required. We revised the manuscript accordingly to correct the wording and accurately reflect the use of non-parametric methods, see lines 177-184.

“The finding that not attending CME is associated with higher practice scores (p = .001) should be explored for confounding factors such as seniority or specialty mix. A multivariable analysis, for example a linear regression of practice score adjusting for age, gender, years in practice, specialty, role, and guideline awareness, would allow testing whether the association remains independent.”

Response: We have conducted a multiple linear regression analysis to examine the association between CME attendance and practice scores while adjusting for potential confounding variables, including age, gender, years in practice, specialty, role, and guideline awareness. The results indicate that practice score remains significantly associated with CME attendance, independent of CME attendance (β = 1.368; 95% CI: 0.738–1.998; p < 0.001) (see Table 8).

“3) A third set of issues involves sampling, external validity, and representativeness. Recruitment via social media with convenience sampling is pragmatic but may bias toward younger, digitally engaged clinicians, which is reflected in the fact that 59 percent of respondents were aged 24–34. The authors should discuss selection bias and coverage error explicitly, and temper the claim of being a nationwide survey accordingly. If possible, they should compare the regional and specialty distribution of their sample to national physician statistics. The computation of a minimum sample size based on the total physician population is appropriate in theory, but the actual recruitment strategy did not use probability sampling. The sampling frame, number of invitations sent, view/click/participation rates, and eligibility screening should be clarified, so that readers can better judge nonresponse bias.”

Response: We thank the reviewer for this helpful comment. The manuscript has been revised to clarify the sampling frame, inclusion/exclusion criteria, and recruitment strategy. We have also expanded the discussion of selection bias, coverage error, and limitations of convenience sampling via social media. The nationwide claim has been tempered accordingly, and the unavailability of national distribution data has been noted, see pages 23-24.

“4) The definition of “general practitioner” versus “general physician” is another source of confusion. The manuscript defines a general practitioner as post-internship without training and a general physician as working in a specialized field without training. This classification may be unusual internationally and risks confusion. The categories should be aligned with standard nomenclature or more clearly justified with examples. Alternatively, the categories could be merged or renamed.”

Response: We acknowledge that our original term “general physician” may be misleading internationally, as it could be interpreted to mean a trained internist or specialist in general medicine. To avoid confusion, we have revised the terminology in the manuscript. We now use the term “Physician without specialty training” to describe doctors who, after completing internship, are working in hospital or outpatient clinical services (e.g., internal medicine, general surgery, orthopaedics, etc.) but who have not entered or completed a specialty residency program. This term more accurately reflects their role in the Saudi healthcare system while minimizing the risk of misinterpretation by an international readership.

Accordingly, the revised definitions in the Methods section now read:

General Practitioner (GP): A medical doctor who has graduated, completed the one-year internship, and is not enrolled in any specialty training (residency) program. These physicians typically work in primary healthcare centers.

Physician without specialty training: A medical doctor who has similarly completed medical school and internship, who is practicing in a clinical field such as internal medicine, surgery, or orthopaedics, but who has not entered or completed a formal specialty training (residency) program.

We believe this clarification addresses the reviewer’s concern and provides greater consistency with both local Saudi classification and international understanding, see lines 113-119.

“5) Another issue is the alignment of item keying with guidelines. The Introduction and Methods cite the ACR 2020 guideline, but the knowledge item on AH thresholds appears keyed as >7 mg/dL for men and >6 mg/dL for women, while the manuscript later acknowledges varying definitions across guidelines, such as 6.8 mg/dL for both sexes or 6 mg/dL suggested in other sources. The authors should state explicitly which definition they adopted a priori for item keying, acknowledge heterogeneity across guidelines, and consider a sensitivity analysis re-scoring the knowledge item using alternative accepted thresholds, reporting how knowledge categories change. For dietary items, several “correct answers” appear mis-keyed. These should be verified and corrected, and all affected analyses re-run with updated tables, figures, and text. A supplement should be provided with the full questionnaire and answer key.”

Response: We thank the reviewer for this comment. We have now addressed the reviewer comment in the method section, see lines 156-163 and supplementary table 1.

“6) Interpretation and causal language also need correction. The current text sometimes implies causal effects, such as CME “increases practice” or “knowledge leads to improved practice,” which are not warranted in a cross-sectional design. The language should be reframed in terms of associations. Multivariable and interaction analyses could provide a richer understanding of observed patterns, such as specialty by seniority.”

Response: We thank the reviewer for the comment. All sentences have been revised to describe associations rather than effects.

“7)The abstract and results should report effect sizes, such as mean differences with confidence intervals, alongside p-values for key comparisons,”

Response: Done

“and avoid over-emphasizing percentage labels (“poor,” “moderate,” “good”) without justification for cutoffs.”

Response: Done, and sensitivity analysis with different cutoffs was performed to assess the findings robustness.

“The methods should cite the source questionnaire from Qassim more fully and describe which items were retained, modified, or added, and why. The pilot sample size, criteria for item revision, and reliability coefficients should be included.”

Response: Done

“Typographical errors such as “Ne York” should be corrected,”

Response: Done

We appreciate your careful attention to detail. The identified typographical error has been corrected accordingly.

“assumption checks added, and the analysis strategy made consistent.”

Response: The normality assumptions were checked, and the results of skewness and kurtosis, additionally with Kolmogorov-Smirnov test

“In results tables, 95 percent confidence intervals should be added, reference groups clearly labeled, and the direction of differences indicated. After multiple testing correction, only those comparisons that remain significant should be marked as such.”

Response: Done

 “The dietary items in particular should be re-tabulated after re-keying, since as currently written the narrative risks inadvertently endorsing misaligned dietary advice.”

Response: We thank the reviewer for highlighting this issue. We have carefully verified all dietary items against current guideline recommendations and confirmed that the keys are correct.

“The discussion should expand the limitations, including non-probability sampling, lack of response rate, potential misclassification from brief instruments, possible differential item functioning across specialties, and heterogeneity of AH definitions. Comparisons with prior studies should add contextual explanations such as differences in policy environment, CME availability, or specialty mix, and avoid selective citation.”

Response: Done

“8) Regarding ethics and transparency, IRB approval and consent are reported, which is positive, but more detail should be added about data confidentiality procedures, for example whether IP addresses were handled or duplicate submissions prevented, which is standard in online surveys. The data availability statement that “data are included in the article” is insufficient. The anonymized dataset and the full questionnaire with the key should be deposited in a public repository such as Zenodo or OSF and the accession link provided. Funding and conflicts of interest statements appear clear.”

Response: Done

“9) The references cited are relevant but the background could be better updated and balanced. Recent high-quality guidelines on AH thresholds, treat-to-target serum urate, initiation of ULT during flares, and dietary recommendations should be included, and the item keying aligned accordingly. Some references are dated or regional; including more recent consensus sources would strengthen the foundation. Presentation and clarity can also be improved by fixing typographical and formatting errors, ensuring all tables and figures are correctly numbered and cross-referenced, and providing supplements with the full instrument, scoring guide, pilot details, re-keyed diet items with citations, and a statistical analysis plan that matches the executed analyses.”

Response: Done

Round 2

Reviewer 1 Report

Comments and Suggestions for Authors

Authors have addressed all comments

Author Response

Dear Reviewer 1,

Thank you for confirming that we have addressed all of your comments.

Reviewer 2 Report

Comments and Suggestions for Authors

The authors have made the necessary revisions. Finally, the main text requires a thorough review. Why is there a lot of boldface? If this type of writing is unnecessary, using regular spelling would be more appropriate. This work is acceptable, but I recommend making the revisions I mentioned above during the proofreading phase. 

Author Response

Dear Reviewer 2,

Thank you, we have now removed all bold font, other formatting issues will be handled upon the production phase of the manuscript.

Reviewer 3 Report

Comments and Suggestions for Authors

Dear authors, congratulations for your improved revision. Some minor issues:

  • The three-item knowledge section remains weak despite acknowledgment in the limitations; it might still be viewed as conceptually thin.
  • Some redundancy persists in the Discussion, which could be further condensed.
  • The claim of “nationwide” scope is still a bit overstated given convenience sampling.

Author Response

Dear Reviewer 3,

Comment: “Dear authors, congratulations for your improved revision. Some minor issues:

  • Comment: The three-item knowledge section remains weak despite acknowledgment in the limitations; it might still be viewed as conceptually thin.
  • Reply: Thank you for this comment, we have now highlighted this point further in the limitations section and added recommendations for future research to address this point as the following “the knowledge section included only three items, which may limit the ability to fully assess physicians’ understanding of AH and is considered conceptually thin. Therefore, future studies should increase the number of items used to examine the knowledge in order to enhance the internal consistency and content coverage.”.
  • Comment: Some redundancy persists in the Discussion, which could be further condensed.
  • Reply: Thank you for this comment, we have now checked the discussion section and removed duplicated test to make sure that the text is further condensed.
  • Comment: The claim of “nationwide” scope is still a bit overstated given convenience sampling.”
  • Reply: Thank you, we have now addressed this point and removed the word “nationwide” from the study title.